# Turning to religion as a mediator of the relationship between hopelessness and job satisfaction during the COVID-19 pandemic among individuals representing the uniformed services or working in professions of public trust in Poland

**Krzysztof Jurek**[1]*, **Iwona Niewiadomska**[2], **Leon Szot**[3]

**1** Department of Sociology of Culture, Religion and Social Participation, John Paul II Catholic University of Lublin, Lublin, Poland, **2** Department of Social Psychoprevention, John Paul II Catholic University of Lublin, Lublin, Poland, **3** Faculty of Social Sciences, The Pontifical University of John Paul II in Cracow, Cracow, Poland

\* kjurek@interia.eu

**Data Availability Statement:** All relevant data presented in this study are publicly available in the

## Abstract

### Introduction

During the COVID-19 pandemic individuals performing uniformed service or working in a profession of public trust were particularly exposed to chronic stress. The exposure to stress contributes to a decrease in quality of life across various domains, including professional performance. The perceived mental difficulties can lead to a feeling of hopelessness which, in turn can generate a decrease in job satisfaction. Religiosity is a factor which, in stress-inducing conditions, not only stops the spiral of perceived resource losses but also triggers gains in the resources possessed.

### Aim

The aim of the study was to assess the preference for positive religious coping strategies, namely turning to religion as a mediator for the relationship between perceived hopelessness and job satisfaction in the individuals declaring religiosity during the COVID-19 pandemic. The analysis has been performed based on the Conservation of Resources theory (COR).

### Methods

The study encompassed 238 individuals representing the uniformed services or working in professions of public trust in Poland. The Inventory for Measuring Coping with Stress (MINI-COPE) and the Beck Hopelessness Scale (BHS) were used in the research.

Zenodo repository (https://doi.org/10.5281/zenodo.8088147).

**Funding:** The author(s) received no specific funding for this work.

**Competing interests:** The authors have declared that no competing interests exist.

## Results

The mediating role of turning to religion in relationship between perceived hopelessness and job satisfaction was confirmed only in the group of women. The relationship found in this group indicates that perceived hopelessness is alleviated by turning to religion, which simultaneously leads to an increase in job satisfaction.

## Conclusion

The obtained results prove that counselling should be standard practice after potentially traumatic events in the workplace; moreover, emotional and/or instrumental support should be offered along with spiritual one.

## Introduction

Professions of public trust are those that consist in performing duties of a special nature in the context of the pursuit of public interest. In the Polish legal system, professions of public trust currently include also medical and related professions, such as physicians, paramedics, nurses, and midwives. As regards uniformed service, it is associated with performing duties as part of formal units created by state authorities to carry out basic or important tasks for the state and its citizens. The tasks assigned to uniformed services result in the strict subordination component being much more developed there than in labor law relations—partly because an individual in this kind of service must be ready to sacrifice their health or even their life to protect values and goods directly related to the security of the state and its citizens.

The term "uniformed service" is usually used with reference to agencies and institutions such as the Internal Security Agency, the Foreign Intelligence Agency, the Central Anticorruption Bureau, the State Fire Service, the Police, the Border Guard, the Prison Service, the Military Intelligence Service, the Military Counterintelligence Service, the Marshal's Guard, and the State Protection Service [1].

### Hopelessness and job satisfaction

During the COVID-19 pandemic, individuals performing uniformed service or working in a profession of public trust are particularly exposed to chronic stress. Such exposure is related to prolonged negative emotional tension that may lead to negative consequences such as psychosomatic problems, increased use of psychoactive substances (e.g., alcohol, tranquillizers), aggressive behavior (e.g., suicidal ideation or suicide attempts, uncontrolled outbursts of anger, hostility towards others), states of anxiety, apathy, post-traumatic stress syndrome (PTSD), and depression (including hopelessness as one of the symptoms) [2].

The consequences of stress, in turn translate into a decrease in quality of life across various domains, including professional performance. Therefore, the objective of the research was the assessment of factors contributing to the cessation of the spiral of losses generated by the individuals' functioning under long-term job related stress. Reports found in the literature suggest that religiosity is a factor that both stops the spiral of perceived resource losses and triggers gains in the resources possessed in stress-inducing conditions [3–7].

In the context of the patterns, it is reasonable to investigate if a preference for positive religious coping strategies in the individuals reporting religiosity is a mediator of the relationship between perceived hopelessness and job satisfaction in conditions of prolonged occupational

stress (performing uniformed service or working as medical professionals during the COVID-19 pandemic). Hopelessness is defined as a mental state based on a system of cognitive schemas that reflect a mostly negative attitude towards the future and towards oneself in a future time perspective. This is accompanied by a belief about the unchangeability of negative situations due to the fact that a person will never be able to achieve the goals they consider important, to solve their problems, and to correct what is right or important [8].

In the context of life changes (e.g., losses or illnesses), hopelessness is therefore an experience contrary to hope which can be defined as a positive motivational state (other than optimism) that involves distinct thinking about the ways of achieving one's goals and/or solving one's problems through personal engagement. What may serve as an example is the mediating function of hope of success in the relationships between the remaining components of psychological capital (self-efficacy, optimism, and resilience) and quality of life in working retirees [9]. In this article, hope means perseverance toward the achievement of set goals and taking actions that increase the possibility of success. What is also characteristic is the simultaneous thinking about goals and ways of achieving them, such as seeking alternative options to blocked realization paths that stimulate energy and increase the sense of being in control of events instead of experiencing helplessness. Therefore, this concept should be distinguished from optimism that is defined in the literature as a positive attitude regarding obtaining success currently or in the future.

The experience of hopelessness can result from a momentary or prolonged experience of failures and/or difficulties in generating solutions to the problems that arise, which leads the person to believe that there is no way out of the current situation. Therefore, the experience of hopelessness can be said to result from non-constructive problem-solving that contributes to the perception of the world and oneself in terms of hopelessness [10]. Hopelessness is sometimes referred to as helplessness or learned helplessness when it results from repeated failures or experiences that the person has interpreted as failures [11].

## Conservation of resources theory and hopelessness

The feeling of hopelessness can also be interpreted in the light of Stevan Hobfoll's Conservation of Resources (COR) theory, which allows for predicting the psychological consequences of the problems experienced. The COR theory explains two phenomena—experiencing stress and constructing psychological resilience to different conditions. Its main assumptions include the following four principles: primacy of loss, resource investment, gain paradox and desperation. The COR theory assumes that stress in people occurs when: a) there is a threat of loss of key resources for them; b) important resources have been lost; c) there is no possibility of gaining important resources following the effort to obtain them. The primacy of loss principle means that resource loss is disproportionately more salient than resource gain. The essence of the principle of resource investment is that people must invest resources in order to protect against resource loss, recover from losses, and gain resources in order to develop psychological resilience. The paradox principle implies that resource gain increases in salience in the context of resource loss. When the circumstances of resource loss are high, resource gains become more important, in order to reduce the stress experienced and / or its negative consequences. The principle of desperation implies that exhaustion or outstretching of resources leads to defensive behavior. The preference for defensive/non-adaptive behavior is not accidental. People enter a defensive mode to preserve the maximum amount of resources in reserve in case further losses need to be countered. The above mentioned processes are characteristic of the entire resource structures. Resources do not exist individually but "travel" in caravans, for both individuals and organizations. In stressful situations, representing a variety of challenges, it is

possible to use different configurations of resources. A person can reach for each of them individually, and/or can use selected combinations of them [12, 13].

Analyzing the mechanisms postulated by the COR theory, one might conclude that hopelessness results from resource losses which make up short-term and/or long-term loss spirals or cycles. The level of hopelessness is probably the highest in those individuals who have the lowest level of resources and/or the smallest possibilities of regaining them. This conclusion is consistent with the results of studies in which it was found that in both short (6 months) and long time periods (12 months) large resource losses were significant risk factors for mental disorders or difficulties, including depression, anxiety, and PTSD [13–16].

The mechanism presented above is supported by research results, showing that hope as an important personal resource is a significant indicator of mental and physical health [17, 18]. By contrast, hopelessness—attesting to the loss of this personal resource—often leads to negative outcomes in human functioning. Among other things, it may lead to maladaptive behaviors (e.g., acts of aggression towards others, substance use, suicide attempts), somatic disorders, and/or certain mental disorders (mainly depression) [19–21].

However, the relations between hopelessness, depressive symptoms, and suicidal acts have not been clearly determined so far. Research on depression revealed that hopelessness was an important mediator between other symptoms of this disorder and suicidal actions, whereas in other studies it was found that hopelessness could be an independent risk factor for suicidal behaviors [22, 23]. Still other authors report that hopelessness favors suicidal ideation to a greater degree than other depressive symptoms—partly due to the fact that the lack of hope largely reflects helplessness in the context of strong mental suffering, which increases immediately or within a short time after a traumatic situation [24].

An important dimension of general quality of life is job satisfaction, professional activity being one of the dominant areas of human functioning [25, 26]. Job satisfaction is defined as a person's positive or negative attitude towards their professional activity (incl. the degree to which the results achieved meet the expectations) and their workplace [27, 28]. Thera are various elements that generate satisfaction in the work environment, such as satisfaction with remuneration, opportunities to engage in creative activities, autonomy, work organization and/or working conditions, interpersonal relations, or opportunities to develop and continue learning [29, 30]. Job satisfaction is one of the most often measured variables in the work environment because a lack of it often leads to passiveness, low organizational engagement, low efficiency, absence from work, job burnout, and/or resignation from work [31].

The studies reported in the literature found significant relationships between mental health and job satisfaction [32]. However, the relationship between the variables mentioned should be noted not to be one-directional. On the one hand, it has been observed that low job satisfaction (e.g., as a result of organizational changes, lack of promotion opportunities, excessively high demands, little freedom in decision making, perceived uncertainty of employment, or lack of support from co-workers) can contribute to the development of mental difficulties (incl. feelings of depression, anxiety, and job burnout). On the other hand, perceived mental difficulties can lead to a decrease in job satisfaction [33]. One of the factors to be thought of as decreasing job satisfaction is hopelessness. Looking at the relationship between hopelessness and job satisfaction through the lens of the COR theory, it is possible to conclude that the experience of short- and long-term cycles of resource losses that give rise to the feeling of hopelessness significantly co-occurs with a decline in job satisfaction. Based on the mechanisms embodied in the principle of desperation, it can be assumed that the sense of hopelessness is the result of experienced resource losses that are not stopped and/or balanced by the resilient functions of capital gains. In the long term, a negative cognitive triad may occur, which

includes the following beliefs: 1) "I am worthless," 2) "the world is an unfair place," and 3) "I will always experience failure in the future." The cognitive errors outlined generate adaptive difficulties and defensive behaviors, including to lower job satisfaction [34].

It is therefore reasonable to ask if there are factors that can inhibit the relations between hopelessness and job satisfaction decrease. The literature suggests that one of such factors may be religiosity, which refers to beliefs, practices, and rituals associated with Transcendence (the Higher Power, the Ultimate Truth) and rooted in the established tradition that can be practiced in private and/or in public with a group of people sharing common beliefs and practices concerning the sacred [35]. In the article, religiosity is not considered synonymously with spirituality. The spiritual sphere includes human experiences that give meaningfulness, purpose and high value to one's existence, e.g. feeling in harmony with the world, ethical sensitivity, altruism, inner freedom, gratitude, opposition to evil, an ability to forgive. On the other hand, religiosity refers to the internal mental processes involved in experiencing a certain relationship with God, treated as a reality that exists outside the visible world. Religiosity can be inferred from such elements as religious awareness and feelings, religious decisions made, ties to the community, religious practices, morality, religious experiences and forms of religious life. It should be emphasized that both constructs—spirituality and religiosity—are complex and multidimensional in nature. For this reason, they may overlap, or they may be interrelated but not identical. The gender differences are marked in the experience of religiosity. Namely, in women, religious experiences are more likely to be spontaneous, intense, with the presence of a variety of feelings and the perception of God in terms of a loving father and friend. In contrast, in the religious experiences of men, there is more often an element of rationality directed at the desire to know and understand the Transcendent, combining religious feelings with intellectual processes, emphasizing the importance of the rules of religious life, perceiving God as a ruler and guardian of the law. Despite the aforementioned differences in religious experiences, conclusive results in the intensity of religiosity in men and women have not been found in the research [36, 37].

There are several arguments in favor of the above conclusion.

## Religiosity and depression syndrome

Religiosity is an important dimension of human functioning regardless of nationality and culture. This is evidenced by the information on 13,000 ethnolinguistic groups from 238 countries, 5,000 cities, and 3,000 provinces, collected in the World Christian Database. According to the published report, atheists account for less than 0.01% of the population in 24 countries and less than 0.1% population in 100 countries (in which such data are available), while at the same time they constitute more than 5% of the population only in 9 countries: Cuba, Latvia, Uruguay, Vietnam, China, Mongolia, Kazakhstan, Sweden, and North Korea [38]. The Gallup Poll conducted towards the end of 2011 in the United States also revealed that for 55% of Americans religion was very important in life, for 26% it was moderately important, and for 19% it was not important [39].

Religiosity contributes to reducing the level of the depression syndrome, in which hopelessness is one of the dominant symptoms. This is supported by the results of a meta-analysis of 444 studies conducted between 1962 and 2011. In more than 60% of the analyzed research reports, religiosity was concluded to be conducive to a spontaneous decrease in the level of depressive disorders and/or to a decrease in the level of depression as a result of an intervention. The results support the conclusion that religious beliefs and practices can help people to better cope with stressful life situations, give new meanings to traumatic circumstances, and provide hope and/or social support.

Only in a small number of studies (6%) did religious beliefs lead to an increase in the sense of guilt and/or to discouragement as a result of failure to meet the requirements stemming from high religious standards [40]. Similar findings were presented based on a meta-analysis of 23 studies devoted to relations between religiosity and depressive symptoms. Namely, in 19 out of the 23 published reports significant relationships were found between a decrease in the level of depressive symptoms and religious functioning (measured by indicators such as religious affiliation, general religious commitment, organizational religious commitment, prayer, private religious commitment, religious motivation, and religious beliefs). In longitudinal studies researchers concluded that religious commitment could protect a person from the emergence and/or persistence of depressive disorders. In some articles it was also reported that in more religious populations the prevalence of depression was lower [41]. For example, fewer negative attitudes and depressive disorders were found in Catholics than in people professing other religions [42, 43].

## Religious coping strategies and stress

Research results indicate a strict relationship between religiosity and psychological well-being [44, 45]. At the same time, an increase in spiritual well-being is conducive to a decrease in hopelessness and/or depressive symptoms [46, 47]. The results of the study also indicate that spiritual well-being (which includes both religious and psychosocial aspects) significantly correlates with the experience of hopelessness and perceived self-efficacy. An increase in spiritual well-being leads to a decrease in perceived hopelessness and an increase in self-efficacy [48].

People often use religious coping strategies in situations of stress. In the literature, authors formulate the conclusion that positive religious coping serves as a common and potentially effective coping strategy in traumatic life situations, including somatic diseases and/or mental health difficulties [49–51]. The use of turning to religion as a strategy for coping with stress is a sign of activity in the spiritual dimension that reflects a secure relationship and a sense of a spiritual bond with the Higher Power [52]. Additionally, positive religious coping can generate meaning in life, strengthen the sense of belonging (i.e., reduce loneliness), alleviate stress, protect a sense of control, induce and/or maintain hope, and strengthen self-worth [53–55].

The results of a meta-analysis conducted on the results of 49 studies support the conclusion that there is a moderate positive correlation between positive religious coping strategy and the positive outcomes of stress (incl. growth in the domains of spirituality, self-esteem, social relations, and quality of life) and that there is a negative relationship between turning to religion in problem situations and a decrease in perceived mental health difficulties (e.g., PTSD, anxiety, or depression) [56]. In the context of the COR theory, the constructive outcomes of positive religious coping strategies may argue that personal resources in the form of turning to religion help a person cope with problems by enabling him or her to build a relatively coherent picture of reality in which even stress-inducing traumatic events are meaningful [57]. The importance of the processes of reducing stress and/or its negative consequences are the relationships that occur between the turn to religion and the personal resources that create subjective capital. This capital is associated with: 1) the belief that it is necessary to make a cautious effort to succeed in difficult tasks (self-efficacy); 2) a positive attitude regarding obtaining success in the current and/or future timeframe (optimism); 3) perseverance toward achieving the goals set despite challenges and setbacks (hope for success) [58].

This kind of pattern has been observed during the COVID-19 pandemic. Namely, positive religious coping made it possible during this period to constructively cope with various traumatic events—leading to outcomes that included a reduction of stress, the alleviation of negative emotions, and constructive coping with mourning [59–61].

## Positive religious strategies for coping and religiosity

What is important is the link between positive religious strategies for coping with stress and religiosity reflecting the person's rootedness in beliefs, practices, and rituals established by tradition. This kind of link indicates that the preference for turning to religion in difficult situations concerns the established (religious) beliefs, which not only serve as the basis for the formation of attitudes but also constitute an important motivating factor in circumstances associated with making important life decisions [62, 63].

The above link between positive religious coping strategies and religiosity is supported by research results. It has been found, among other things, that a high level of problems, being affected by natural disasters, and/or the experience of diseases motivate people all over the world to use religious coping strategies and to increase their religious commitment [64]. It can therefore be concluded that religiosity is a personal resource that is important for using positive religious coping strategies in problem situations.

It is, to a great extent, the integration of these factors that makes turning to religion not only a protective factor for physical and mental health in the process of coping with stress but also an important predictor and determinant of the person's functioning in various domains of life. It can be noted, for instance, that intrinsic religious orientation moderates the relationship between occupational burnout and job satisfaction by reducing the negative effect of perceived burnout on the level of satisfaction with one's work. Extrinsic religious orientation, by contrast, does not perform the moderating function in the analyzed respect [65].

The relations between a preference for positive religious strategies of coping with stress and maintaining satisfaction with one's work by means of resources based on religiosity may stem from factors such as: (1) preferring individual values that are shaped and/or supported by religion; (2) social influence highlighting the value of work, present in religious teachings and communities; (3) generalized sense of high quality of life and/or meaning in life stemming from religious faith [66–68].

Based on the data found in the literature, the following three hypotheses were formulated concerning the individuals researched performing uniformed service or working in a profession of public trust during the COVID-19 pandemic:

*Hypothesis 1*: There is a negative relationship between hopelessness and job satisfaction in conditions of prolonged occupational stress; namely an increase in hopelessness co-occurs with a decrease in job satisfaction.

*Hypothesis 2*: For people who declare themselves religious a preference for positive religious coping strategies is a mediator of the relationship between perceived hopelessness and job satisfaction in conditions of prolonged occupational stress. The mediating functions of the turn to religion for the basic relationship support mechanisms implied by the COR theory. The high intensity of variables constituting the caravan of subjective resources (including a turn to religion, self-efficacy or optimism) should significantly generate changes in the negative cognitive triad (accompanying feelings of hopelessness) toward more positive views of the self (increased self-esteem), world, and future (hope for success), which should consequently contribute to increased job satisfaction under long-term stress.

*Hypothesis 3*: Gender is a factor moderating the mediation effect of turning to religion (i.e., the use of positive religious coping strategies) on the relationship between hopelessness and job satisfaction in conditions of prolonged occupational stress.

## Materials and methods

### Participants and procedure

The study was conducted according to the guidelines of the Declaration of Helsinki, and written-approved by the Ethics Committee of the Institute of Sociological Sciences of the John Paul II Catholic University of Lublin (protocol code: 18/DKE/NS2022).

The study included 238 individuals representing the uniformed services or working in professions of public trust: 155 health professionals, 45 firefighters, 27 soldiers and 11 policemen. The subjects were directly involved in the care of patients with COVID-19. The participants had critical roles and responsibilities during the pandemic. The recruitment of the respondents was carried out on two levels. Firstly, 3 single-purpose hospitals dedicated to treating patients with suspected or confirmed COVID-19 (so called 'COVID—hospital', 21 hospitals in Poland) were randomly selected. Secondly, invitations to selected hospitals to participate in the study were sent. The link for the questionnaire was shared via e-mail and social media to potential participants of the study. Finally, each participant received a questionnaire form and an informed consent form. Participation in the study was voluntary and anonymous. A total of 411 questionnaires were collected, of which 238 were correctly completed. The average age was 42.47 ± 12.91 years old (ranging from 20 to 71 years). Seventy-three point one per cent of the participants were women. The characteristics of the participants are included in Table 1.

### Measures

**Hopelessness.** The Beck Hopelessness Scale (BHS) is composed of 20 dichotomous "true/false" items that aimed to assess three major aspects of hopelessness: feelings about the future, loss of motivation, and expectations. Total scores were created by first reverse-coding nine items (items 1, 3, 5, 6, 8, 10, 13, 15, 19) and then summing the item scores. Higher total scores indicate greater hopelessness (range 0–20). The Polish version of the BHS has been translated and validated by Oleś and Juros [69]. The internal reliability coefficient is reasonably high (0.87). In this study, the internal consistency of the BHS was adequate (0.83).

**Table 1. Participant demographics.**

| Variable | | Percentage or Mean [N = 238] |
|---|---|---|
| Gender | Men | 26.9 [N = 64] |
| | Women | 73.1 [N = 174] |
| Age | | 42.5 [SD = 12.9] |
| Seniority | | 18.2 [SD = 12.9] |
| Uniformed services or working in professions of public trust | Health professionals | 65.1 [N = 155] |
| | Firefighters | 18.9 [N = 45] |
| | Soldiers | 11.3 [N = 27] |
| | Policemen | 4.7 [N = 11] |
| Education | Vocational | 2.5 [N = 6] |
| | Secondary | 36.6 [N = 87] |
| | Higher | 60.9 [N = 145] |
| Marital status | Single/ divorced/ widowed | 39.7 [N = 94] |
| | Married | 60.3 [N = 144] |
| Place of residence | Urban area | 68.9 [N = 164] |
| | Rural area | 31.1 [N = 74] |

## Turning to religion

The Polish version of the Mini-COPE [70] inventory was used to measure "turning to religion" as a coping strategy. It is a shortened version of the Multimodal Inventory for Measurement of Coping with Stress-COPE (by Carver, Scheier, and Weintraub) and measures coping in terms of disposition. It consists of 28 items that are part of 14 strategies for coping with stress, including active coping, planning, positive revalidation, acceptance, sense of humor, turning to religion, seeking emotional support, seeking instrumental support, taking care of something else, denial, discharge, use of psychoactive substances, cessation of activities, and self-blaming. There are two theorems for each strategy. The tested respondent refers to each statement by marking one possible answer on a four-point Likert scale where 0 means "I almost never do so" and 3 means "I almost always do so". The obtained psychometric properties are satisfactory. The half-reliability for 14 scales is 0.86 (Guttman's index 0.87). In this study, only one stress coping strategy—turning to religion was analyzed. The domain consists of two items "I've been trying to find comfort in my religion or spiritual beliefs" and "I've been praying or meditating". The internal consistency of the strategy was high (0.87).

**Job satisfaction.** The respondents were asked to rate their degree of job satisfaction on a five-point Likert scale, where 0 means "very dissatisfied" and 5 means "very satisfied" including a neutral response choice.

## Statistical methods

The participants' demographic characteristics were analyzed and descriptive statistics were computed using PS IMAGO. The simple path analysis models were estimated using PROCESS software. Significance was determined at the $p < .05$ level. We applied a bootstrapping procedure with 5,000 bootstrap samples to test the significance of the total and indirect effects and the differences in these effects across the levels of the moderator variables. The 95% confidence intervals for the coefficients calculated using the bootstrapping method were considered statistically significant if they did not include zero.

## Mediation effect

In the first model, the direct effect of hopelessness on job satisfaction (path c) was analyzed. The total effect comprises path c', which represents the direct effect between hopelessness and job satisfaction with the coping strategy of turning to religion controlled for and the indirect effect (path ab) of hopelessness on job satisfaction via turning to religion. A mediation analysis was performed to find out if the relationship between hopelessness and job satisfaction was mediated by turning to religion. This model was used to check if there was a significant indirect effect between hopelessness and job satisfaction. If after entering the mediator variable the direct effect (path c') is lower but still significant, partial mediation is found. Full mediation means that the direct effect (path c') is no longer significant after the mediator variable has been entered (Fig 1).

## Moderated mediation

In the second model, moderated mediation was analyzed to refer to the mediation effect that differs depending on the level of the moderator variable. The moderator variable was gender. Testing the mediation effects in each subgroup (women and men) leads to a biased estimate of the parameter and to low statistical power. The estimation of the parameters (total, indirect, and direct effects) was performed for the analyzed model by integrating the moderation and mediation methods. Gender moderated the relationship between the use of the turning to religion strategy and job satisfaction (path b) (Fig 2).

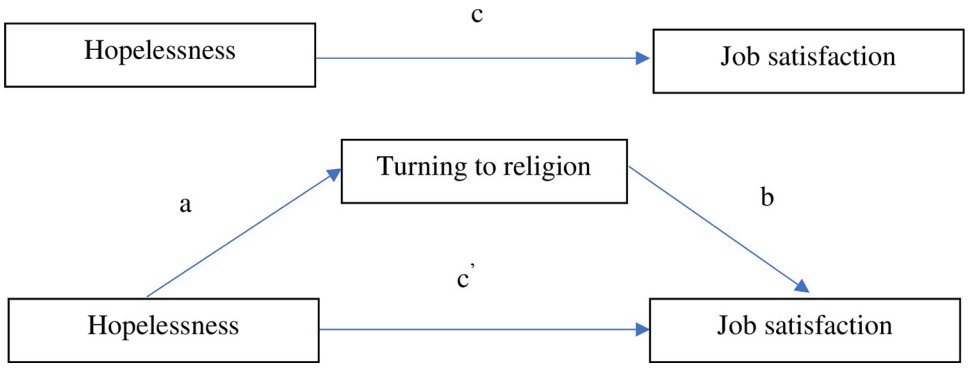

**Fig 1. General mediation model.**

## Results

The job satisfaction had positive correlation with turning to religion. The turning to religion had negative correlation with hopelessness. The hopelessness had negative correlation with job satisfaction (Table 2).

Fig 3 shows the results of the simple mediation analysis (PROCESS, model 4) with turning to religion as a mediator in the relationship between hopelessness and job satisfaction. The total effect of hopelessness on job satisfaction was significant, $B$ = -0.0520 ($SE$ = 0.0129), 95% BootCI [LLCI = -0.0773, ULCI = -0.0266]. The direct effect of hopelessness on job satisfaction was significant too, $B$ = -0.0453 ($SE$ = 0.0137), 95% BootCI [LLCI = -0.0723, ULCI = -0.0183]. However, the indirect effect of hopelessness on job satisfaction was not significant, $B$ = 0.1113 ($SE$ = 0.0802), 95% BootCI [LLCI = -0.0468, ULCI = 0.2694].

Fig 4 shows the results of the moderated mediation analysis (PROCESS, Model 14) with gender as a moderator variable and turning to religion as a mediator in the relationship between hopelessness and job satisfaction. The index of moderated mediation was significant, $B$ = 0.0182 ($SE$ = 0.0091), 95% BootCI [LLCI = 0.0009, ULCI = 0.0371] indicating that the indirect effect of turn to religion was moderated. There was a significant interaction effect of gender, $B$ = -0.3039 ($SE$ = 0.1536), 95% BootCI [LLCI = -0.6069, ULCI = -0.0010]. Inspection of the conditional indirect effects revealed that there was a significant mediation effect of turning to religion in the case of women ($B$ = 0.3238, $SE$ = 0.1245, 95% BootCI [LLCI = 0.0782, ULCI = 0.5693]), but not in the case of men ($B$ = 0.0198, $SE$ = 0.0988, 95% BootCI [LLCI = -0.1752, ULCI = 0.2148]).

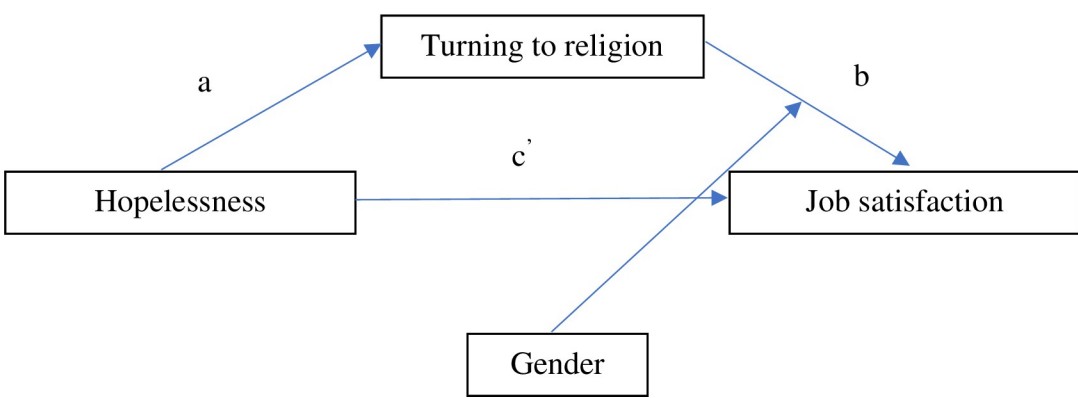

**Fig 2. General moderated mediation model.**

**Table 2. Pearson r correlations between the study variables.**

|  | 1 | 2 | 3 |
|---|---|---|---|
| 1. Job satisfaction | - | 0.189** | -0.281*** |
| 2. Turning to religion |  | - | -0.351*** |
| 3. Hopelessness |  |  | - |
| **M (SD)** | 3.5 (0.8) | 1.4 (0.9) | 7.9 (4.2) |

* p < 0.05
** p < 0.01

## Discussion

The research enabled the formulation of several important conclusions. Firstly, there is a negative relationship between hopelessness and job satisfaction under prolonged occupational stress (in the case of individuals performing uniformed service or working in a profession of public trust during the COVID-19 pandemic). The relationship consists in the co-occurrence of an increase in perceived hopelessness with a decrease in job satisfaction and is consistent with research findings in most developed countries where the mental health difficulties experienced by employees (mainly depressive and/or anxiety symptoms) lead to a decrease in job satisfaction and/or to disturbances in job performance (e.g., sick leaves or long-term incapacity to work), leading to negative outcomes in employees, employers, and society.

The relationship found in the research can be explained based on COR theory postulating that a person's basic aspiration is to gain, maintain, support, and protect the resources that constitute an important or central value for them. In this perspective, hopelessness largely reflects the losses in personal resources that, on the one hand, result from the experience of failures and, on the other, lead to the belief that there is no way out of the current situation.

The COR theory also assumes that not only specific resources but above all their combinations or fusions perform regulatory functions [71, 72]. It can therefore be suggested that the decrease in job satisfaction which accompanies hopelessness is largely caused by losses in resources referred to in the literature as psychological capital. This term signifies a positive psychological state of individual development that is generated by: (1) a belief that one should

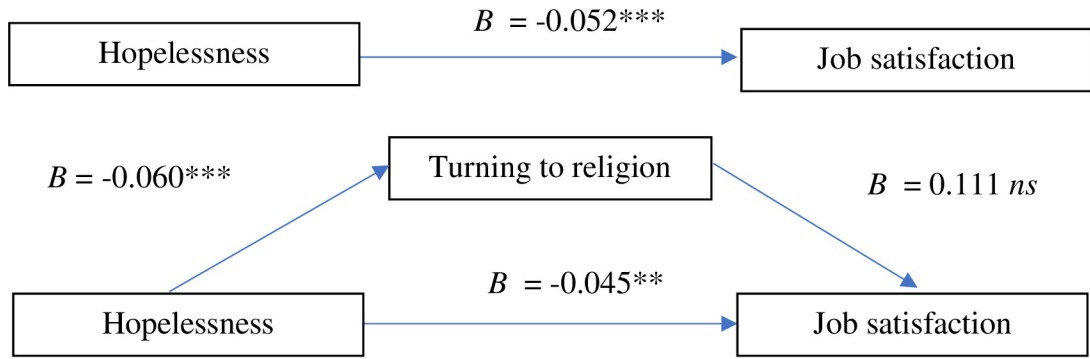

*Unstandardized coefficients*
*** < 0.001;** < 0.01*

**Fig 3. Model of relationships between hopelessness, turning to religion, and job satisfaction.** Unstandardized coefficients ***
< 0.001;** < 0.01.

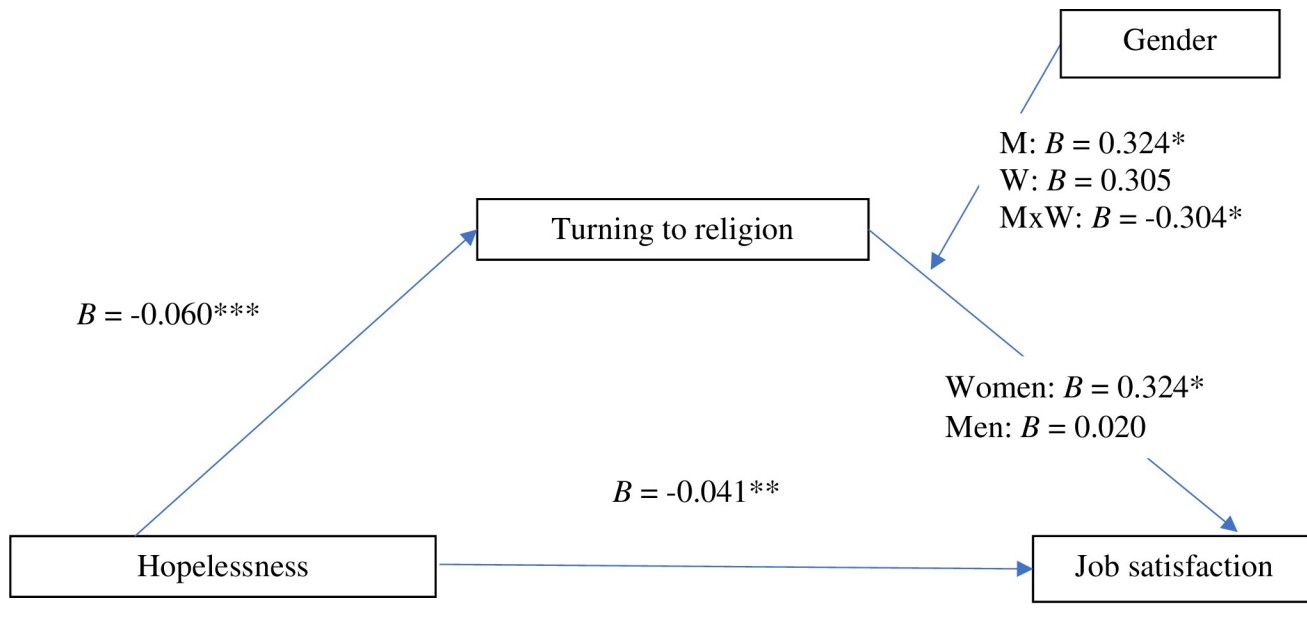

**Fig 4. The moderation effects of gender in the relationships between hopelessness, turning to religion, and job satisfaction.** Unstandardized coefficients *** < 0.001; ** < 0.01; * < 0.05.

make careful efforts to succeed in difficult tasks (self-efficacy); (2) a positive attitude regarding the achievement of success at present and/or in the future (optimism); (3) resistance to problems and/or easily/flexibly regaining balance when one has been affected by the negative outcomes of the difficulties experienced (resilience); (4) perseverance in striving to achieve one's goals and directing one's activities towards such efforts that increase the likelihood of succeeding (hope of success).

The second conclusion from the study is that in religious people (i.e., ones reporting religiosity) a preference for positive religious coping strategies under prolonged occupational stress does not act as a mediator of the relationship between perceived hopelessness and job satisfaction. This conclusion is based on the results indicating that turning to religion when individual is under chronic stress reduces the level of perceived hopelessness. There is, however, no significant relationship between the level of positive religious coping strategies and the level of job satisfaction. The presented relationship confirms the results of research in which it was found that individuals experiencing a sense of hopelessness develop new patterns of hope in situations when they are able to find meaning in their life—for instance, thanks to the preferred values [73]. This kind of pattern has also been observed in analyses performed during the COVID-19 pandemic. They revealed that religiously committed individuals were higher than self-reported non-believers in all dimensions of spiritual well-being, including religious beliefs, meaning in life, and quality of interpersonal relationships. The results obtained in this study also confirm the pattern reported in the literature, in which the use of positive religious coping strategies in problem situations protects the person against the negative effects of stress such as depressive symptoms, anxiety, post-traumatic stress, and frequent alcohol and/or substance use [74].

The third conclusion based on the presented results is that gender is a factor which moderates the mediating functions of turning to religion in the relationship between hopelessness

and job satisfaction in conditions of prolonged stress. Namely, in the group of religious women performing uniformed service or working in a profession of public trust during the COVID-19 pandemic the use of the positive religious coping strategy was a mediator of the relationship between perceived hopelessness and the experience of job satisfaction. The relationship found in this group indicates that perceived hopelessness is alleviated by turning to religion, which at the same time leads to an increase in job satisfaction. An attempt to explain this pattern has been presented in the context of several mechanisms discussed in Hobfoll's COR theory. The first of these consists in the fact that initiating behaviors which amount to positive religious coping strategies (e.g., positive religious reappraisal, reliance on a secure relationship with the merciful God, seeking spiritual support, religious practices, and prayer) is a kind of perceived resource gain, based on religious personal resources. The second mechanism is that perceived resource gains in the form of turning to religion contribute to stopping the cycle of losses in resources (incl. personal resources) that the person subjectively interprets in terms of hopelessness. In the third mechanism, perceived resource gains as a result of using positive religious coping strategies leads to generating a cycle/spiral of resource gains related to occupational functioning, which in turn lead to an increase in job satisfaction [75].

The fourth conclusion is a practical one. The presented results support the recommendation consistent with the suggestions found in the literature that it should be standard practice to offer consultation with a specialist after potentially traumatic events in the workplace not only in the form of emotional and/or instrumental support but also in the form of spiritual help [76].

In the context of the relationships between hopelessness, positive religious coping, and job satisfaction, it is also important to take gender differences into account. This is shown by the results of studies that revealed several significant patterns in this respect. As far as hopelessness is concerned, it was found, for instance, that in the first phase of the COVID-19 pandemic women perceived a higher level of mental health difficulties than men [43]. Similar patterns were found in research based on the COR theory. Namely, a high level of depressive symptoms was found in women who experienced highly intense resource loss spirals and who at the same time had a small amount of resources. Moreover, it was observed that, to a significant degree, gender determined the strategies applied to sustain or regain hope [77]. As regards the religiosity domain, research results show that both in Catholic societies and in Islamic ones women are characterized by a higher level of religiosity and spiritual well-being than men [78–80].

During the COVID-19 pandemic women not only showed higher religious commitment compared to men but also more often used positive religious strategies for coping with stress. Additionally, it was found that religiously committed women (including those who preferred positive religious coping strategies in difficult situations) were less prone to depressive symptoms and hopelessness [81].

## Conclusion

In some people the COVID-19 pandemic may lead to mental health difficulties (such as post-traumatic stress disorder, depressive symptoms, and hopelessness) while in others it may lead to post-traumatic growth (PTG) and/or to an increase in quality of life [82, 83]. Differences in the way of responding to prolonged stress provide a reason to look for factors that could contribute to an improvement in perceived quality of life during the COVID-19 pandemic.

Positive religious coping strategies based on long-term religious commitment may be factors protecting a person from the negative effects of stress, especially when he or she experiences hopelessness while struggling with difficulties. In such circumstances, activity in the spiritual sphere is a special way of confronting problems which attests to trust in God and shows that one has put one's fate into the hands of the Higher Power [84].

## Limitations

A cross-sectional study was conducted. Studies of this type can identify potential correlations, associations and relationships between variables. The study was only exploratory; thus, cause-and-effect relationships are difficult to identify. Another limitation of this study is related to the data collection method. The data collection used self-reported results, which may cause bias. The participants are often biased when they report on their own experiences. Subjects may make the more socially acceptable answer rather than being truthful or may not be able to assess themselves accurately.

In addition, the researchers were in contact with hospitals that served as healthcare settings solely for COVID-19 patients (both suspected and confirmed ones) during the pandemic (in Poland there were 21 hospitals that were called 'COVID—hospitals'). The link for the questionnaire was shared via e-mail and social media to potential participants of the study. During the pandemic, it was the only way to contact the respondents. Online-only design strategies are comparatively inexpensive and they can be carried out relatively quickly. However, this method of recruitment has several disadvantages. Online-only approaches completely exclude people who are unable to participate in online surveys because they lack the Internet access or the necessary skills to use the Internet. This type of sampling does not provide information about respondents who actually decide to participate or not to participate in the study and how many of those who received the invitation took part in the study.

Future studies should include several elements. First, they should include the identification of factors that act as moderators of the relationship between hopelessness and job satisfaction in individuals working under prolonged/chronic stress (e.g., age, working conditions, or type of work). Second, they should be aimed at identifying the variables that act as mediators in this relationship—such as various ways of coping with long-term stress. Third, they should deepen the analyses to verify the result indicating that gender is a moderator for the mediation effect of turning to religion (a preference for positive religious coping strategies) on the relationship between the level of perceived hopelessness and job satisfaction.

## Author Contributions

**Conceptualization:** Krzysztof Jurek, Iwona Niewiadomska.

**Data curation:** Iwona Niewiadomska.

**Formal analysis:** Krzysztof Jurek.

**Methodology:** Krzysztof Jurek, Iwona Niewiadomska.

**Project administration:** Krzysztof Jurek, Iwona Niewiadomska.

**Supervision:** Iwona Niewiadomska, Leon Szot.

**Validation:** Krzysztof Jurek, Iwona Niewiadomska.

**Writing – original draft:** Krzysztof Jurek, Iwona Niewiadomska.

**Writing – review & editing:** Krzysztof Jurek, Iwona Niewiadomska, Leon Szot.

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
