## [Decision Letter · Decision Letter 0]

15 May 2023

PONE-D-23-02708Turning to Religion as a Mediator of the Relationship Between Hopelessness and Job Satisfaction During the COVID-19 Pandemic Among Individuals Representing the Uniformed Services or Working in Professions of Public Trust in PolandPLOS ONE

Dear Dr. Jurek,

Thank you for submitting your manuscript to PLOS ONE. After careful consideration, we feel that it has merit but does not fully meet PLOS ONE’s publication criteria as it currently stands. Therefore, we invite you to submit a revised version of the manuscript that addresses the points raised during the review process.

We look forward to receiving your revised manuscript.

Kind regards,

Piotr Janusz Mamcarz, Doctor

Academic Editor

PLOS ONE

Journal Requirements:

a) Did participants provide their written or verbal informed consent to participate in this study?

5. We note you have included a table to which you do not refer in the text of your manuscript. Please ensure that you refer to Table 1 and 2 in your text; if accepted, production will need this reference to link the reader to the Table.

Reviewers' comments:

Reviewer's Responses to Questions

**Comments to the Author**

1. Is the manuscript technically sound, and do the data support the conclusions?

Reviewer #1: Partly

Reviewer #2: Yes

2. Has the statistical analysis been performed appropriately and rigorously? 

Reviewer #1: Yes

Reviewer #2: Yes

3. Have the authors made all data underlying the findings in their manuscript fully available?

Reviewer #1: No

Reviewer #2: Yes

4. Is the manuscript presented in an intelligible fashion and written in standard English?

Reviewer #1: Yes

Reviewer #2: Yes

5. Review Comments to the Author

Reviewer #1: The authors investigated the mediating role of positive religious coping strategies in the relationship between perceived hopelessness and job satisfaction. They administered a self-report questionnaire to 238 individuals representing the uniformed services or working in professions of public trust in Poland. Results showed that the hypothesized mediating pathway was confirmed only for women. All in all, I think that the idea behind the study is interesting, with potentially relevant implications for researcher and practitioners. At the same time, I believe that the study has some drawbacks, concerning the theoretical framework as well as the result section. Detailed comments are reported below.

***

- p.11, "Analyzing the mechanisms postulated by the COR theory". I think that it would be useful for readers to briefly review at this stage the main assumption of the COR theory, at least those essential to the current study.

- p.13, " the experience of short- and long-term cycles of resource losses that give rise to the feeling of hopelessness significantly co-occurs with a decline in job satisfaction". Do the authors mean that these phenomena occur because of common-cause, or that they influence each other? Based on my reading or the paper and in the light of the COR, I thought that hopelessness could lead to (less) job satisfaction. The authors should clarify this point.

- p.14, "Research results also show that spiritual well-being [...] performs mediating functions between hopelessness and self-efficacy, decreasing the former while increasing the latter". This sentence is not clear, in my opinion. If spiritual well-being is a mediator, how can it decrease the former (i.e., the independent variable)? The authors should clarify this point.

- p.15, "personal resources in the form of turning to religion help a person cope with problems by enabling him or her to build a relatively coherent picture of reality in which even stress inducing traumatic events are meaningful". This is an interesting point, but I think that it should be expanded. What are the resources - according to the definition provided by the COR - that turning to religion should replenish/preserve (e.g., positive emotions, psychological/physical energy)?

- p.17, "In religious people (i.e., in ones reporting religiosity) a preference for positive religious coping strategies is a mediator of the relationship between perceived hopelessness and job satisfaction in conditions of prolonged occupational stress". I think that the reasoning behind this hypothesis should be more robust. When I first read the manuscript, I thought that positive religious coping strategies could be a moderator - rather than a mediator - in the relationship between perceived hopelessness and job satisfaction. Stated differently, the relationship between perceived hopelessness and job satisfaction could differ, depending if religious individuals adopt more/less religious coping strategies.

- p.17, "In conditions of prolonged occupational stress (performing uniformed service or working in a profession of public trust during the COVID-19 pandemic)". Although both occupations should be characterized by prolonged occupational stress, these sources of stress are not necessarily the same across occupations. Why were these occupational groups not considered separately (e.g., by dummy coding in the models)?

- p.17, "The index of moderated mediation was significant". What do the authors mean by index of moderated mediation? Also, it would be useful to see in the text the different values of the regression coefficient for "turning to religion" to "job satisfaction" for women and men, together with bootstrap-based confidence intervals. Finally, it seems to me that results in the figure/text should be double-checked (e.g., possible different values of coefficient for men?).

- p.26, Limitations. To my understanding, this is a cross-sectional, single-method study. If so, this should be acknowledged in the limitations section of the manuscript. In this regard, did the authors checked possible presence of common method bias?

Reviewer #2: Comments to the Author

Thank you for the opportunity to review the manuscript entitled “Turning to Religion as a Mediator of the Relationship Between Hopelessness and Job Satisfaction During the COVID-19 Pandemic Among Individuals Representing the Uniformed Services or Working in Professions of Public Trust in Poland”.

General Comments

The study concerns an important issue in the field of public health during the COVID-19 pandemic. Especially in times of pandemic, studies on coping strategies make an important contribution providing practically relevant knowledge for a better handling of pandemic-related stressors. Due to the fact that too little attention is paid to research on religiosity as a protective factor in the field of public health, I am very happy to review this study.

The manuscript is well written, the methods and statistical analyses are appropriate. The results seem consistent and plausible. The discussion is clear and coherent, the conclusions are adequately drawn from the demonstrated data. I have only some minor points that should be considered in the revision:

Introduction:

- Could you shortly describe what is the difference between hope and optimism.

- Please provide a definition for religiosity and for spirituality because these constructs mean not the same, although they are often used synonymously. Please check throughout the manuscript if both constructs are used correctly.

- Hypothesis 2 -> what exactly do you mean with “religious people”? Did you also measure the degree of religiosity, e.g. the intrinsic religiosity? It could be postulated that intrinsic religious persons would have a preference for positive religious coping strategies, while it would not be the case for extrinsic religious individuals.

- As your third hypothesis concerns gender, could you please describe some findings on gender and religiosity.

Methods:

Participants and procedure: What do you mean with “correctly completed” questionnaires? You refer to the response rate as the amount of “correctly completed” questionnaires out of the 411 (238/411). This is not correct. Your response rate is 411/x. Is it possible for you to define the number of the total sample if all respondents have participated when you have used also social media for the recruitment? Please add this aspect to the limitations and describe the implications.

- Table 1 -> If possible please add the information for the religious affiliation of the participants. Besides, please add the absolute frequencies and the standard deviations, where appropriate.

- The last sentence above Table 1 should be: “Seventy-three point one percent of the participants were women.” Please correct.

- Hopelessness: The Polish version of the BHS … (not Polsih)

- Please add the information on Cronbach´s Alpha obtained in the present study for the BHS and the Mini-COPE.

References:

- You list too many references (this number is adequate for a review, not for a research article), please reduce the number of references.

- There are some references with XXX -> please provide the correct references!

I hope my comments are helpful to the authors and to the editorial team.

6. PLOS authors have the option to publish the peer review history of their article (what does this mean?). If published, this will include your full peer review and any attached files.

Reviewer #1: No

Reviewer #2: No

---

## [Author Response · Author response to Decision Letter 0]

27 Jun 2023

Reviewer 1

Dear Sir or Madam,

We kindly request submission of the revised manuscript "Turning to Religion as a Mediator of the Relationship Between Hopelessness and Job Satisfaction During the COVID-19 Pandemic Among Individuals Representing the Uniformed Services or Working in Professions of Public Trust in Poland” (Manuscript ID: PONE-D-23-02708) for further consideration by the PLOS ONE. We appreciate all the comments provided by the Reviewer. We have taken them into account and due to the extensive knowledge of the Reviewer we have learnt a lot. We have applied changes to the manuscript according to the valuable comments. The improvements are marked in blue in the manuscript.

Point 1: p.11, "Analyzing the mechanisms postulated by the COR theory". I think that it would be useful for readers to briefly review at this stage the main assumption of the COR theory, at least those essential to the current study.

Response 1: We have added information regarding the main assumptions of the COR theory.

The COR theory explains two phenomena - experiencing stress and constructing psychological resilience to different conditions. Its main assumptions include the following four principles: primacy of loss, resource investment, gain paradox and desperation. The COR theory assumes that stress in people occurs when: a) there is a threat of loss of key resources for them; b) important resources have been lost; c) there is no possibility of gaining important resources following the effort to obtain them. The primacy of loss principle means that resource loss is disproportionately more salient than resource gain. The essence of the principle of resource investment is that people must invest resources in order to protect against resource loss, recover from losses, and gain resources in order to develop psychological resilience. The paradox principle implies that resource gain increases in salience in the context of resource loss. When the circumstances of resource loss are high, resource gains become more important, in order to reduce the stress experienced and / or its negative consequences. The principle of desperation implies that exhaustion or outstretching of resources leads to defensive behavior. The preference for defensive/non-adaptive behavior is not accidental. People enter a defensive mode to preserve the maximum amount of resources in reserve in case further losses need to be countered. The above mentioned processes are characteristic of the entire resource structures. Resources do not exist individually but "travel" in caravans, for both individuals and organizations. In stressful situations, representing a variety of challenges, it is possible to use different configurations of resources. A person can reach for each of them individually, and/or can use selected combinations of them.

Point 2: - p.13, " the experience of short- and long-term cycles of resource losses that give rise to the feeling of hopelessness significantly co-occurs with a decline in job satisfaction". Do the authors mean that these phenomena occur because of common-cause, or that they influence each other? Based on my reading or the paper and in the light of the COR, I thought that hopelessness could lead to (less) job satisfaction. The authors should clarify this point.

Response 2: We have added information indicating correlations between feelings of hopelessness and a decline in job satisfaction.

Based on the mechanisms embodied in the principle of desperation, it can be assumed that the sense of hopelessness is the result of experienced resource losses that are not stopped and/or balanced by the resilient functions of capital gains. In the long term, a negative cognitive triad may occur, which includes the following beliefs: 1) "I am worthless," 2) "the world is an unfair place," and 3) "I will always experience failure in the future." The cognitive errors outlined generate adaptive difficulties and defensive behaviors, including to lower job satisfaction.

Point 3: p.14, "Research results also show that spiritual well-being [...] performs mediating functions between hopelessness and self-efficacy, decreasing the former while increasing the latter". This sentence is not clear, in my opinion. If spiritual well-being is a mediator, how can it decrease the former (i.e., the independent variable)? The authors should clarify this point.

Response 3: We have removed the sentence "Research results also show that spiritual well-being (comprising both religious and psychosocial aspects) performs mediating functions between hopelessness and self-efficacy, decreasing the former while increasing the latter". We replaced it with the sentence: 

The results of the study also indicate that spiritual well-being (which includes both religious and psychosocial aspects) significantly correlates with the experience of hopelessness and perceived self-efficacy. An increase in spiritual well-being leads to a decrease in perceived hopelessness and an increase in self-efficacy.

 Point 4: p.15, "personal resources in the form of turning to religion help a person cope with problems by enabling him or her to build a relatively coherent picture of reality in which even stress inducing traumatic events are meaningful". This is an interesting point, but I think that it should be expanded. What are the resources - according to the definition provided by the COR - that turning to religion should replenish/preserve (e.g., positive emotions, psychological/physical energy)?

Response 4: We have supplemented the article with the following part.

The importance of the processes of reducing stress and/or its negative consequences are the relationships that occur between the turn to religion and the personal resources that create subjective capital. This capital is associated with: 1) the belief that it is necessary to make a cautious effort to succeed in difficult tasks (self-efficacy); 2) a positive attitude regarding obtaining success in the current and/or future timeframe (optimism); 3) perseverance toward achieving the goals set despite challenges and setbacks (hope for success).

Point 5: p.17, "In religious people (i.e., in ones reporting religiosity) a preference for positive religious coping strategies is a mediator of the relationship between perceived hopelessness and job satisfaction in conditions of prolonged occupational stress". I think that the reasoning behind this hypothesis should be more robust. When I first read the manuscript, I thought that positive religious coping strategies could be a moderator - rather than a mediator - in the relationship between perceived hopelessness and job satisfaction. Stated differently, the relationship between perceived hopelessness and job satisfaction could differ, depending if religious individuals adopt more/less religious coping strategies.

Response 5: We have supplemented the article with the following part.

The mediating functions of the turn to religion for the basic relationship support mechanisms implied by the COR theory. The high intensity of variables constituting the caravan of subjective resources (including a turn to religion, self-efficacy or optimism) should significantly generate changes in the negative cognitive triad (accompanying feelings of hopelessness) toward more positive views of the self (increased self-esteem), world, and future (hope for success), which should consequently contribute to increased job satisfaction under long-term stress.

Point 6: p.17, "The index of moderated mediation was significant". What do the authors mean by index of moderated mediation? Also, it would be useful to see in the text the different values of the regression coefficient for "turning to religion" to "job satisfaction" for women and men, together with bootstrap-based confidence intervals. Finally, it seems to me that results in the figure/text should be double-checked (e.g., possible different values of coefficient for men?).

Response 6: In the article there were errors in the presentation of the results (in the graphs the results are correct, the errors did not affect the interpretation of the results). We have corrected that. We have reviewed the results in the text. We do appreciate the reviewer’s meticulousness and we have applied the valuable comment.

The index of moderated mediation quantifies the relationship between the indirect effect and a moderator.

Point 7: p.26, Limitations. To my understanding, this is a cross-sectional, single-method study. If so, this should be acknowledged in the limitations section of the manuscript. In this regard, did the authors checked possible presence of common method bias?

Response 7: We have supplemented the limitations section with the following part.

We conducted cross-sectional – study. Studies of this type can identify potential correlations, associations and relationships between variables. Our study was only exploratory, thus it is difficult to identify cause-and-effect relationships. Another limitation of this study is related to the data collection method. The data collection used self-reported results, which may cause bias. The participants are often biased when they report on their own experiences. Subjects may make the more socially acceptable answer rather than being truthful or may not be able to assess themselves accurately.

We used the Monte Carlo Power Analysis for Indirect Effects application developed by Schoemann, Boulton and Short. 

Point 8: p.17, "In conditions of prolonged occupational stress (performing uniformed service or working in a profession of public trust during the COVID-19 pandemic)". Although both occupations should be characterized by prolonged occupational stress, these sources of stress are not necessarily the same across occupations. Why were these occupational groups not considered separately (e.g., by dummy coding in the models)?

Response 8: We strongly agree with the statement that the sources of stress are not necessarily the same across occupations. However, from the perspective of the COR theory, the sources of stress are not important. We assumed that the inclusion criterion would be performing uniformed service or working in a profession of public trust during the COVID-19 pandemic. To be honest, the Reviewer's proposal seems very interesting, this solution may bring unexpected results.

Reviewer 2

Dear Sir or Madam,

Thank you for all comments. We appreciate your suggestions, they allowed us to look at the discussed problems from a wider perspective. In our opinion, they are also valuable in relation to further research that we are conducting. We strongly believe that the suggested corrections have resulted in an improvement of our manuscript. Below we present our responses to the suggestions and recommendations.

Point 1: Could you shortly describe what is the difference between hope and optimism.

Response 1: We have described the difference between hope and optimism.

In this article, hope means perseverance toward the achievement of set goals and taking actions that increase the possibility of success. What is also characteristic is the simultaneous thinking about goals and about the ways of achieving them–such as seeking alternative options to blocked realization paths that stimulate energy and increase the sense of being in control of events instead of helplessness. Therefore, this concept should be distinguished from optimism, which is defined in the literature as a positive attitude regarding obtaining success currently or in the future. 

Point 2: Please provide a definition for religiosity and for spirituality because these constructs mean not the same, although they are often used synonymously. Please check throughout the manuscript if both constructs are used correctly.

Response 2: We have provided definition for religiosity and for spirituality.

In the article, religiosity is not considered synonymously with spirituality. The spiritual sphere includes human experiences that give meaningfulness, purpose and high value to one's existence, e.g. feeling in harmony with the world, ethical sensitivity, altruism, inner freedom, gratitude, opposition to evil, an ability to forgive. On the other hand, religiosity refers to the internal mental processes involved in experiencing a certain relationship with God, treated as a reality that exists outside the visible world. Religiosity can be inferred from such elements as religious awareness and feelings, religious decisions made, ties to the community, religious practices, morality, religious experiences and forms of religious life. It should be emphasized that both constructs - spirituality and religiosity - are complex and multidimensional in nature. For this reason, they may overlap, or they may be interrelated but not identical.

Point 3: Hypothesis 2 -> what exactly do you mean with “religious people”? Did you also measure the degree of religiosity, e.g. the intrinsic religiosity? It could be postulated that intrinsic religious persons would have a preference for positive religious coping strategies, while it would not be the case for extrinsic religious individuals.

Response 3: We have revised the formulation of the hypothesis.

Point 4: As your third hypothesis concerns gender, could you please describe some findings on gender and religiosity.

Response 2: We have provided some findings on gender and religiosity.

The gender differences are marked in the experience of religiosity. Namely, in women, religious experiences are more likely to be spontaneous, intense, with the presence of a variety of feelings and the perception of God in terms of a loving father and friend. In contrast, in the religious experiences of men, there is more often an element of rationality directed at the desire to know and understand the Transcendent, combining religious feelings with intellectual processes, emphasizing the importance of the rules of religious life, perceiving God as a ruler and guardian of the law. Despite the aforementioned differences in religious experiences, conclusive results in the intensity of religiosity in men and women have not been found in the research.

Point 5: What do you mean with “correctly completed” questionnaires? You refer to the response rate as the amount of “correctly completed” questionnaires out of the 411 (238/411). This is not correct. Your response rate is 411/x. Is it possible for you to define the number of the total sample if all respondents have participated when you have used also social media for the recruitment? Please add this aspect to the limitations and describe the implications.

Response 5: We have removed information on response rate. We have added aspects of the recruitment to the limitations. 

We were in contact with hospitals that served as healthcare settings solely for COVID-19 patients (both suspected and confirmed ones) during the pandemic (in Poland there were 21 hospitals that were called ‘COVID—hospitals’). The link for the questionnaire was shared via e-mail and social media to potential participants of the study. During the pandemic, it was the only way to contact the respondents. Online-only design strategies are comparatively inexpensive and they can be carried out relatively quickly. However, this method of recruitment has several disadvantages. Online-only approaches completely exclude people who are unable to participate in online surveys, because they lack the Internet access or the necessary skills to use the Internet. this type of sampling does not provide information about respondents who actually decide to participate or not to participate in the study and how many of those who received the invitation took part in the study.

Point 6: If possible please add the information for the religious affiliation of the participants. Besides, please add the absolute frequencies and the standard deviations, where appropriate.

Response 6: Unfortunately, we did not collect information about religious affiliation of the participants. We have added the absolute frequencies and the standard deviations.

Point 7: The last sentence above Table 1 should be: “Seventy-three point one percent of the participants were women.” Please correct.

Response 7: We have corrected.

Point 8: Hopelessness: The Polish version of the BHS … (not Polsih). 

We have corrected.

Please add the information on Cronbach´s Alpha obtained in the present study for the BHS and the Mini-COPE.

Response 8: We have added information on Cronbach´s Alpha obtained in the present study:

In this study, the internal consistency of the BHS was adequate (0.83). 

The internal consistency of the strategy was high (0.90). 

Point 9: You list too many references (this number is adequate for a review, not for a research article), please reduce the number of references. 

Response 9: We have removed several items from references

Point 10: There are some references with XXX -> please provide the correct references!

Response 10: The symbol “XXX” has been placed in the review version of the article on purpose so as to avoid the provision of names and surnames of the coauthors of the manuscript. The references section will be supplemented with full information after the process of the review.

---

## [Decision Letter · Decision Letter 1]

19 Jul 2023

PONE-D-23-02708R1Turning to Religion as a Mediator of the Relationship Between Hopelessness and Job Satisfaction During the COVID-19 Pandemic Among Individuals Representing the Uniformed Services or Working in Professions of Public Trust in PolandPLOS ONE

Dear Dr. Jurek,

Thank you for submitting your manuscript to PLOS ONE. After careful consideration, we feel that it has merit but does not fully meet PLOS ONE’s publication criteria as it currently stands. Therefore, we invite you to submit a revised version of the manuscript that addresses the points raised during the review process.

We look forward to receiving your revised manuscript.

Kind regards,

Piotr Janusz Mamcarz, Doctor

Academic Editor

PLOS ONE

Journal Requirements:

Reviewers' comments:

Reviewer's Responses to Questions

**Comments to the Author**

1. If the authors have adequately addressed your comments raised in a previous round of review and you feel that this manuscript is now acceptable for publication, you may indicate that here to bypass the “Comments to the Author” section, enter your conflict of interest statement in the “Confidential to Editor” section, and submit your "Accept" recommendation.

Reviewer #1: (No Response)

Reviewer #2: All comments have been addressed

2. Is the manuscript technically sound, and do the data support the conclusions?

Reviewer #1: Yes

Reviewer #2: Yes

3. Has the statistical analysis been performed appropriately and rigorously? 

Reviewer #1: Yes

Reviewer #2: Yes

4. Have the authors made all data underlying the findings in their manuscript fully available?

Reviewer #1: No

Reviewer #2: Yes

5. Is the manuscript presented in an intelligible fashion and written in standard English?

Reviewer #1: Yes

Reviewer #2: Yes

6. Review Comments to the Author

Reviewer #1: I thank the authors for addressing my concerns. I believe that the revised manuscript might be suitable for publication in the journal, once some revisions are made. A detailed description is reported below.

***

- Point 1. I thank the authors provided a detailed description of the COR Theory, which will be useful for reader unfamiliar with its principles. However, since the COR developed over the years, I would add also a recent reference, such as for example Hobfoll et al., 2018, doi: 10.1146/annurev-orgpsych-032117-104640

- Point 2. I thank the authors for including a detailed description but, again, I think that a citation would provide a useful link for further readings.

- Point 6. I thank the authors for correcting the results reported in the paper. However, I think that a double check would still be necessary.

(6.1) For example, results in Figure 3 are internally consistent, given that the sum of direct and indirect effect in the lower part (−0.351*0.102−0.245) are equal to the total effect in the upper part (−0.281), as well as with the correlation in Table 2 (again, −0.281). However, to my understanding, the standardized beta of hopelessness on turning to religion in Figure 3 (−0.351) should be equal to a bivariate correlation coefficient, which is however -0.171 in Table 2. (6.2) Second, as far as I understand, standardized regression coefficients are reported in Figure 3, but unstandardized regression coefficients are shown in Figure 4, which makes results difficult to compare. What is the rationale behind this choice? (6.3) Similarly, the unstandardized regression coefficient of hopelessness on turning to religion in Figure 4 (−0.600) does not seem consistent with the standardized regression coefficient in Figure 3 (−0.351), but please correct me if I am wrong. (6.4) Finally, please also note that the symbols "-" and "−" are interchangeably used for "minus". Similarly, both "." and "," are used as decimal separator (e.g., Figure 3).

- Point 7. I thank the authors for the thorough discussion of possible limitations concerning the cross-sectional research design. However, I do not understand the reference to the "Monte Carlo Power Analysis for Indirect Effects application developed by Schoemann, Boulton and Short". As far I understand, this procedure is aimed at investigating statistical power, not the presence of common method bias. In this respect, please see a review by Podsakoff et al., 2003 (doi: 10.1037/0021-9010.88.5.879).

- Finally, I noticed some typos, so the manuscript needs a revision of the English form.

Reviewer #2: Thank you very much for the revised version of the manuscript. All my suggested improvements have been adequately implemented! I fully endorse publication of this revised manuscript. I thank all authors for this important scientific contribution and wish them all the best.

7. PLOS authors have the option to publish the peer review history of their article (what does this mean?). If published, this will include your full peer review and any attached files.

Reviewer #1: No

Reviewer #2: **Yes: **PD Dr. Eva Morawa

---

## [Author Response · Author response to Decision Letter 1]

7 Aug 2023

Dear Reviewer 1,

Thank you very much for the Reviewer’s careful reading of our manuscript. The suggested changes have improved the manuscript considerably. We hope that you will find our manuscript acceptable in its current form. All the improvements introduced are presented below.

Point 1: I thank the authors provided a detailed description of the COR Theory, which will be useful for reader unfamiliar with its principles. However, since the COR developed over the years, I would add also a recent reference, such as for example Hobfoll et al., 2018, doi: 10.1146/annurev-orgpsych-032117-104640

Response 1: We have added the indicated reference to the manuscript and bibliography.

Point 2. I thank the authors for including a detailed description but, again, I think that a citation would provide a useful link for further readings.

Response 2: We have added the publication given below to the manuscript and bibliography.

McIntosh CN, Fischer DG. Beck's cognitive triad: One versus three factors. Canadian Journal of Behavioural Science. 2000;32(3): 153–157. doi.org/10.1037/h0087110.

Point 3. I thank the authors for correcting the results reported in the paper. However, I think that a double check would still be necessary. (6.1) For example, results in Figure 3 are internally consistent, given that the sum of direct and indirect effect in the lower part (−0.351*0.102−0.245) are equal to the total effect in the upper part (−0.281), as well as with the correlation in Table 2 (again, −0.281). However, to my understanding, the standardized beta of hopelessness on turning to religion in Figure 3 (−0.351) should be equal to a bivariate correlation coefficient, which is however -0.171 in Table 2. 

Response 3: We do appreciate the Reviever’s careful reading and analyzing our manuscript. We do agree with the Reviewer. Thanks to the Reviewer’s meticulousness we could correct the mistake made in the table. We have made the corrections. 

Point 4. Second, as far as I understand, standardized regression coefficients are reported in Figure 3, but unstandardized regression coefficients are shown in Figure 4, which makes results difficult to compare. What is the rationale behind this choice? 

Response 4: The coefficients in Figure 3 and in the text have been changed into non-standardized ones. We have made it to ensure uniformity. Moreover, we have used the recommendations made in the publication: Fairchild AJ, McQuillin SD. Evaluating mediation and moderation effects in school psychology: a presentation of methods and review of current practice. J Sch Psychol. 2010 Feb;48(1):53-84. doi: 10.1016/j.jsp.2009.09.001. PMID: 20006988; PMCID: PMC5488867.

Point 5: Similarly, the unstandardized regression coefficient of hopelessness on turning to religion in Figure 4 (−0.600) does not seem consistent with the standardized regression coefficient in Figure 3 (−0.351), but please correct me if I am wrong. (6.4) Finally, please also note that the symbols "-" and "−" are interchangeably used for "minus". Similarly, both "." and "," are used as decimal separator (e.g., Figure 3).

Response 5: We have made the corrections.

Point 6. I thank the authors for the thorough discussion of possible limitations concerning the cross-sectional research design. However, I do not understand the reference to the "Monte Carlo Power Analysis for Indirect Effects application developed by Schoemann, Boulton and Short". As far I understand, this procedure is aimed at investigating statistical power, not the presence of common method bias. In this respect, please see a review by Podsakoff et al., 2003 (doi: 10.1037/0021-9010.88.5.879).

Response 6: We are grateful for the literature suggested. We do agree with the Reviewer that this procedure is aimed at investigating statistical power. The reference to the application developed by Schoemann, Boulton and Short is inappropriate and does not provide any useful information. We apologize for the confusion.

Point 7: Finally, I noticed some typos, so the manuscript needs a revision of the English form.

Response 7: The manuscript has been revised by a native speaker.

Dear Reviewer 2,

We are grateful for the Reviewer’s positive evaluation of our work and valuable suggestions that, in our view, helped us to improve the manuscript.

---

## [Editor Report · Decision Letter 2]

24 Aug 2023

Turning to religion as a mediator of the relationship between hopelessness and job satisfaction during the COVID-19 pandemic among individuals representing the uniformed services or working in professions of public trust in Poland

PONE-D-23-02708R2

Dear Dr. Jurek,

Dear authors,

I am pleased to inform you that your manuscript "Turning to religion as a mediator of the relationship between hopelessness and job satisfaction during the COVID-19 pandemic among individuals representing the uniformed services or working in professions of public trust in Poland" has been accepted for publication in our journal. All reviewers' comments were addressed and the manuscript was significantly improved.

The manuscript makes an important contribution by examining the mediating role of positive religious coping between hopelessness and job satisfaction among frontline workers during the pandemic. The use of established theoretical frameworks like the Conservation of Resources theory enhances the significance of the work. The study also highlights pertinent gender differences in the mediating effects of religious coping.

The objectives were clearly specified, and the methodology using validated instruments was sound. The results were thoroughly analyzed and discussed in relation to previous literature. The limitations of the cross-sectional design and self-reported data were appropriately acknowledged. The conclusions and practical implications regarding provision of spiritual support were well-articulated.

In summary, this is a well-conducted study generating insights into how religious coping may promote job satisfaction among distressed frontline workers. The findings have high relevance during public health crises like the COVID-19 pandemic. I believe the work will interest our readers and make a useful addition to the literature.

Therefore, I am pleased to accept your paper for publication. Congratulations on this fine contribution. We look forward to your continued submissions to our journal.

Please let me know if you have any other questions.

Kind regards,

Piotr Janusz Mamcarz, Doctor

Academic Editor

PLOS ONE

---

## [Editor Report · Acceptance letter]

30 Aug 2023

PONE-D-23-02708R2 

Turning to religion as a mediator of the relationship between hopelessness and job satisfaction during the COVID-19 pandemic among individuals representing the uniformed services or working in professions of public trust in Poland 

Dear Dr. Jurek:

I'm pleased to inform you that your manuscript has been deemed suitable for publication in PLOS ONE. Congratulations! Your manuscript is now with our production department. 

Kind regards, 

on behalf of

Dr. Piotr Janusz Mamcarz 

Academic Editor

PLOS ONE